# Influence of SHS Precursor Composition on the Properties of Yttria Powders and Optical Ceramics

**DOI:** 10.3390/ma16010260

**Published:** 2022-12-27

**Authors:** Dmitry Permin, Olga Postnikova, Stanislav Balabanov, Alexander Belyaev, Vitaliy Koshkin, Oleg Timofeev, Jiang Li

**Affiliations:** 1Faculty of Chemistry, N.I. Lobachevsky National Research University, 23 Gagarin Ave., 603022 Nizhny Novgorod, Russia; 2G.G. Devyatykh Institute of Chemistry of High-Purity Substances of the Russian Academy of Sciences, 49 Tropinin Str., 603137 Nizhny Novgorod, Russia; 3Transparent Ceramics Research Center, Shanghai Institute of Ceramics, Chinese Academy of Sciences, Shanghai 201899, China

**Keywords:** self-propagating high-temperature synthesis, yttria, optical ceramics, nanopowders

## Abstract

This study looked at optimizing the composition of precursors for yttria nanopowder glycine–nitrate self-propagating high-temperature synthesis (SHS). Based on thermodynamic studies, six different precursor compositions were selected, including with excesses of either oxidant or fuel. The powders from the precursors of all selected compositions were highly dispersed and had specific surface areas ranging from 22 to 57 m^2^/g. They were consolidated by hot pressing (HP) with lithium–fluoride sintering additive and subsequent hot isostatic pressing (HIP). The 1 mm thick HPed ceramics had transmittance in the range of 74.5% to 80.1% @ 1μm, which was limited by optical inhomogeneity due to incomplete evaporation of the sintering additive. Two-stage HIP significantly improves optical homogeneity of the ceramics. It was shown that an excess of oxidizer in the precursor decreases the powders’ agglomeration degree, which forms large pore clusters in the ceramics.

## 1. Introduction

The development of ceramic technology has opened up new opportunities for the development of laser technology by making it possible to obtain new materials with high optical quality and to advance the development of lasers with unique characteristics, such as high pulse and average power, compactness, ultra-short pulses, etc. [1,2].

One of the most promising laser ceramic materials is yttria, Y_2_O_3_. It has a simple cubic structure with an Ia$\bar{3}$ space group, low phonon energy of 592 cm^–1^ [3], high thermal conductivity k = 12.8 W/(m K) at 300 K [4], provides large absorption and stimulated emission peak cross-sections of active rare-earth (RE) ions as a host material, and has other properties that distinguish it favorably from widely used laser media. Moreover, its significantly lower cost compared to other sesquioxides (e.g., Sc_2_O_3_ and Lu_2_O_3_) is important for practical applications. 

Strong competition during processes of densification and grain growth in RE sesquioxides often forces the use of sintering additives that limit grain growth and/or activate diffusion. In the case of Y_2_O_3_ optical ceramics, zirconium oxide [5,6,7], lanthanum oxide [8,9], or their combination [10] are usually introduced for free vacuum sintering (VS). One of the most obvious disadvantages of sintering additives in laser ceramics is a decrease in the thermal conductivity, which significantly enhances undesirable thermally induced effects [11]. Powder consolidation methods using external pressure, such as hot pressing (HP) [12], spark plasma sintering (SPS) [13], hot isostatic pressing (HIP) [14], or their combination [15], can significantly reduce or eliminate the amount of sintering additives and/or the of use evaporating additives (usually lithium fluoride).

The first hot-pressed Y_2_O_3_ optical ceramics were made by Lefever et al. [16] in 1967 and S. K. Dutta and G. E. Gazza in 1969 [12]. These materials showed good transmittance in the infrared range but had significant scattering in the visible part of the spectrum. In the 1990s, Majima et al. [17] published work on achieving better optical quality in hot-pressed Y_2_O_3_ ceramics in the visible range. Later, Sanghera et al. [18] obtained laser-quality Yb:Y_2_O_3_ ceramics with a lithium fluoride sintering additive by combining HP and HIP. In 2021, Balabanov et al. [19] achieved noticeably higher performance of a Yb:Y_2_O_3_ continuous-wave micro-laser on ceramics fabricated by a similar method. However, this ceramic was inferior in quality to the material obtained by combining VS and HIP, both in terms of the slope efficiency of 64.5% vs. 82.4% [20] and maximum power of 0.735 W vs. 32 W [21]. The most obvious reasons for these differences are residual porosity scattering or/and optical inhomogeneity, which manifests as a deviation of the ceramic transmittance from the theoretical and increases in the short wavelength region of the spectrum. 

Powder characteristics play a crucial role in the formation of residual porosity and optical inhomogeneity; by tailoring the powder characteristics, the quality of Y_2_O_3_ ceramics obtained by HP can be further improved. In [19], glycine-nitrate self-propagating high-temperature synthesis (SHS) of yttria was used. It is known that the parameters of SHS powders are controlled by changing the ratio of the oxidizer and fuel in the reaction mixture [22,23,24,25]. This ratio determines the temperature in the reaction wave and the composition and amount of the substances in the synthesis products.

The purpose of this work is to tailor the Y_2_O_3_ powder properties by changing the oxidizer–fuel ratio in glycine-nitrate self-propagating high-temperature synthesis, as well as to study the relationship between powder properties and characteristics of hot-pressed yttria ceramics.

## 2. Materials and Methods

Yttria Y_2_O_3_ (99.99% Polirit, Russia), nitric acid HNO_3_ (99.9999% Khimreaktiv, Russia), glycine NH_2_CH_2_COOH (99.9% Khimreaktiv, Russia), and lithium fluoride LiF (99.9%) (Khimreaktiv, Russia) were the starting materials for the self-propagating high-temperature synthesis (SHS) precursors.

The range of the SHS precursor’s composition was chosen on the basis of thermodynamic calculations of the adiabatic temperature and equilibrium combustion products using the IVTANTERMO 3.0 software and database [26]. The input data for the thermodynamic calculations were the composition and standard enthalpies of formation (Δ_f_*H*^⦵^) of the starting substances. The Δ_f_*H*^⦵^ values were taken from [27]:

Δ_f_*H*^⦵^ (Y(NO_3_)_3_) = −1070.71 kJ/mol

Δ_f_*H*^⦵^ (NH_2_CH_2_COOH) = −537.225 kJ/mol

The composition of the reaction system was expressed in terms of the mole fraction of yttrium nitrate *φ*$$\varphi =\frac{n\left(Y{\left({\mathrm{NO}}_{3}\right)}_{3}\right)}{n\left(Y{\left({\mathrm{NO}}_{3}\right)}_{3}\right)+n\left({\mathrm{NH}}_{2}{\mathrm{CH}}_{2}\mathrm{COOH}\right)}$$

The enthalpy of formation of the SHS-precursor was determined as the arithmetic mean of the oxidizer and fuel, taking into account their percentages in the starting mixture.

SHS-precursor preparation included dissolving ~10 g of yttria in nitric acid under heating and constant stirring, evaporating the solution to remove excess acid, and dissolving yttrium nitrate in deionized water. The concentration of the solution was determined gravimetrically after calcination of the dry residue at 1200 °C. After that, glycine was added to yttrium nitrate in the calculated amount, and the solution was evaporated at 110 °C. To initiate SHS, the precursor in a quartz flask was placed in a furnace preheated to 500 °C. As a result of the combustion, fine powders were formed.

LiF sintering additive in the form of solution with a concentration of 0.01 g/mL in dilute nitric acid was added to a mixture of yttrium nitrate and glycine, which provided uniform distribution of LiF in the synthesized powders.

X-ray diffraction analysis was performed with a Shimadzu XRD-6000 diffractometer (CuKα radiation *λ* = 1.54178 Å) in the 2θ range of 15°–60°. The scanning step for 2θ was 0.02°, and the exposure time was 3 s. The phase’s parameters were taken from the ICDD database.

The specific surface area (*S*_BET_) of the yttria nanopowder was measured by nitrogen adsorption according to the Brunauer–Emmett–Teller (BET) method using Meta Sorbi-MS equipment. The equivalent particle diameter (*d*_BET_) was calculated assuming their spherical shape according to the following equation: *d*_BET_ = 6/(*ρ* × *S*_BET_), where *ρ* is the theoretical density of the yttria.

Prior to sintering, the powders were compacted in a stainless-steel mold at a pressure of ~10 MPa. The compact was isolated from the punches by graphite paper, placed in a graphite mold (Ø15 mm) and consolidated by hot pressing in vacuum at a maximum temperature of 1500 °C and a uniaxial pressure of 50 MPa using homemade equipment. Heating was carried out by graphite heaters; the residual pressure in the chamber was no more than 10 Pa. The obtained ceramic disks (Ø15 mm) were ground on both sides to a thickness of 1 mm and polished with a diamond suspension. Hot isostatic pressing was performed on the UGL-2000 plant (Russia) in a graphite crucible at 1200 °C and 100 MPa for 20 h, then on the AIP8-30H/WF plant (USA) in an yttria crucible at 1600 °C and 200 MPa for 2 h. The ceramics were then repolished.

The morphology of the powders and the microstructure of the sintered ceramics were studied using an Auriga CrossBeam (Carl Zeiss, Germany) scanning electron microscope at an accelerating beam voltage of EHT = 3 kEV with a secondary electron detector.

The pores present inside the material and the state of birefringence were measured using an AXIOPLAN-2 transmission polarizing microscope (Carl Zeiss, Germany).

The light transmission of the samples was measured using an SF-2000 spectrophotometer (LOMO, Russia) over a wavelength range of 190–1100 nm and an FT-801 FT-IR spectrometer (SIMEX, Russia) over a range of 1.3–20 μm.

## 3. Results and Discussion

### 3.1. Thermodynamic Analysis of the Reaction System Y(NO_3_)_3_–NH_2_CH_2_COOH

An important step in choosing a precursor for self-propagating high-temperature synthesis (SHS) is thermodynamic investigation of reaction systems [28,29]. The precursor composition determines process parameters such as the maximum temperature in the reaction wave, the volume of evolved gases, the intensity of the reaction, etc., which impact the characteristics of the SHS product. Thermodynamic calculations were performed for the considered Y(NO_3_)_3_–NH_2_CH_2_COOH system. At this stage, the effect of lithium fluoride was not taken into account due to its low content in the precursor (1 wt. %). According to the calculations, LiF is a thermodynamically stable compound in the considered system and practically does not participate in reactions (partly due to the lack of reliable data on possible products of the chemical interaction of Y_2_O_3_ and LiF). Its influence is limited by the reduction of the adiabatic temperature due to its heating and evaporation (by approximately 50–100 K depending on the precursor composition). Therefore, it cannot alter the general results of thermodynamic calculations; nevertheless, its effect on the morphology of powders is significant due to changes in the diffusion activity of the components [19].

The calculated adiabatic temperature, mole amount, and volume of gaseous products of combustion in the *φ*Y(NO_3_)_3_–(1-*φ*)NH_2_CH_2_COOH mixture are given in Figure 1a. The maximum adiabatic temperature corresponds to *φ* = 0.375, where the reaction products are in the most thermodynamically stable state: yttria, carbon dioxide, water and nitrogen. The ratio *φ* = 0.375 corresponds to the stoichiometric content of the oxidizer and fuel in the precursor and is expressed by the following chemical reaction: 6Y(NO_3_)_3_ + 10NH_2_CH_2_COOH → 3Y_2_O_3_ + 20CO_2_ + 25Н_2_О + 14N_2_(1)

In similar SHS precursors: yttrium nitrate–yttrium acetate [28], scandium nitrate–glycine [30], lutetium nitrate–lutetium acetate, glycine or citric acid [31], the same dependence is observed. 

The stoichiometric ratio of the oxidizer and fuel is usually used in the SHS of rare earth (RE) sesquioxides (see, for example, [32,33,34]). However, the high temperature of SHS can lead to sintering of Y_2_O_3_ particles and the formation of rigid agglomerates. On the other hand, the temperature cannot be less than some critical value to maintain self-propagation of the reaction. If combustion stops and is replaced by pyrolysis of the precursor, the morphology of the powder changes drastically. Deviation of the SHS reaction from adiabatic conditions and elimination of intermediates from the reaction zone may cause the actual temperature to be different from the calculated one by up to 500 K [28]; thus, for the system under consideration, an adiabatic temperature of about 1700 K is critical.

The large volume of the resulting gaseous products, on the contrary, contributes to the effective dispersion of the formed particles and minimizes their sintering due to the formation of a kind of fluidized bed. According to these parameters, the optimal composition of the precursor corresponds to the lack of an oxidizer and is in the range 0.3 ≤ *φ* < 0.375.

Figure 1b shows the calculated composition of volatile products of the chemical reactions. The condensed phase consists of yttria at all the precursor compositions; therefore, its content is not shown. When the *φ* parameter deviates from stoichiometric to smaller values, carbon monoxide and hydrogen appear in the equilibrium reaction products, increasing the possibility of final product contamination by carbon. With an increase in the oxidizer content, SHS products contain carbon dioxide, nitrogen oxides, and oxygen, which are easily desorbed when heated in a vacuum or in air. Based on this, it is recommended to use precursors with an excess of oxidizer [28,29], which is opposite to the previous conclusion about the positive effect of a lack of an oxidizer for reducing the agglomeration degree of formed particles. However, unlike, for example, powders of yttrium aluminum garnet, in which carbon-containing impurities are not easily removed even by high-temperature calcination [35], there are no data in the literature about the difficulty of removing similar impurities from RE sesquioxides. Such powders clearly contain carbon impurities. After calcination at temperatures of 800–900 °C, they have the color characteristic of the corresponding RE sesquioxide; in the IR spectra, carbon is detected only in the form of carbonates adsorbed on the surface [36,37]. According to the data of simultaneous TG-DSC analysis with mass spectrometry, the maximum emission of CO_2_ from RE sesquioxide powders is observed at temperatures up to about 760 °C, then it decreases noticeably, although it does not become zero [33,36].

Thus, thermodynamic investigation indicates the preferred compositions of SHS precursors of 0.3 ≤ *φ* < 0.375 to reduce the degree of particle agglomeration, but it does not answer whether subsequent annealing can eliminate carbon-containing impurities in powders to an acceptable level for laser ceramics. Therefore, for the experimental study of the SHS precursor composition effect on the properties of yttria powders and optical ceramics, a wider range of compositions of 0.3 ≤ *φ* < 0.5 was chosen.

### 3.2. SHS Y_2_O_3_ Powder Characterization

The combustion of all selected precursor compositions of 0.3 ≤ *φ* ≤ 0.5 took place in self-propagation mode. Y_2_O_3_ fine powders formed as a result of SHS. The obtained yttria powders differed visually depending on *φ* and the presence of LiF sintering additive. Under conditions of oxidizer deficiency, the synthesis product had a gray color, indicating its contamination with carbon. The lithium–fluoride-doped powders were lighter, but, nevertheless, with a large excess of fuel (*φ* = 0.3), they also contained unreacted carbon. Annealing in air at temperatures above 600 °C resulted in a change in the coloration of all powders to white.

Figure 2a shows the X-ray diffraction patterns of the powders obtained from precursors with different ratios of oxidizer and fuel. It can be seen that powders from precursors with *φ* = 0.35, 0.375, and 0.4 possess the highest crystallinity, which is due to the highest combustion temperature during SHS. On the contrary, powders obtained with a large excess of oxidizer or fuel, *φ* = 0.3 and 0.5, are almost amorphous; the temperature and time of the reaction were not sufficient to form crystallites. Additional diffusion constraints for crystallization in powders produced with excess fuel can create combustion intermediates.

Similar precursor compositions with LiF sintering additive show the same differences in diffraction patterns (Figure 2b). However, the crystallinity of the powders is noticeably higher; the diffraction peaks are much more intense and narrower compared to those for powders without LiF. As shown earlier, lithium fluoride is not a separate phase but forms intermediate compounds with yttrium oxide–fluorides and oxofluorides [19]. Due to their low content, these phases are not detected by XRD; only in one sample, *φ* = 0.4, in the region of angles 2θ = 27.5° and 32.2° are there minor peaks, which are close to the most intense peaks of YbF_2.37_ and Y_17_O_14_F_23_ compounds, respectively.

A study of the morphology of SHS powders using scanning electron microscopy also indicates greater crystallinity of lithium–fluoride-doped powders. Figure 3 shows micrographs of Y_2_O_3_ powders obtained with excess fuel and oxidizer, *φ* = 0.3 and *φ* = 0.5, as well as from the stoichiometric mixture, *φ* = 0.375. They have a typical SHS product morphology and are a mixture of nanoparticles combined into large agglomerates of a spongy structure (foams) ranging in size from hundreds of nanometers to tens of micrometers, with a developed surface and low bulk density. The LiF-doped powders obtained from the stoichiometric precursor differ most significantly. The reaction temperature in this case was enough to start changing the morphology of the particles: instead of a lamellar shape, they become rounded; individual particles begin to form from the continuous foam structure.

Quantification of the effect of the oxidizer–fuel ratio on the dispersity of yttrium oxide powders was performed by BET. Figure 4 shows the dependence of the specific surface area (*S*(BET)) and the equivalent diameter (*d*(BET)) of Y_2_O_3_ particles obtained from different precursors. Specific surface area of as-prepared SHS yttria powders varies from 50 to 90 m^2^/g (Figure 4a). Dependence of their dispersity on the composition of the precursor is not observed. Apparently, this is due to the rapid synthesis reactions and the unformed crystalline and porous structure of the powders. As a result of air annealing at 800 °C for 5 h, the residual unreacted components are removed, the Y_2_O_3_ particle size grows, and the specific surface area decreases to ~20–35 m^2^/g (Figure 4b). The minimum values of *S*(BET) are observed for Y_2_O_3_ obtained from the stoichiometric precursor composition, and the dependence of *d*(BET) on *φ* has a form similar to the adiabatic temperature curve in thermodynamic modeling.

The same dependence of *S*(BET) on precursor composition was obtained by Chavan et al. [23] for glycine–nitrate SHS Y_2_O_3_. The authors concluded that at the high temperature of the SHS process, the crystallite size increase and partial sintering of primary particle powders occurs; hence, their *S*(BET) decreases. However, in our case, for example, this dependence is observed for powders calcined at 800 °C, when their size more than doubled compared to the as-prepared one. Considering that finer particles should exhibit greater sintering activity, then, other things being equal, after calcination, the specific surface area of the powders should become approximately equal. Although *S*(BET) difference actually became smaller for all powders, it nevertheless remained significant, on the order of ~50%. This indicates an initial difference in the structure of the powders, which, in turn, is caused by the SHS mechanism. At the front of reaction propagation, the precursor first foams and only then begins combustion, which proceeds with the emission of large amounts of gaseous products [36]. Most probably, the combustion temperature significantly affects the properties of the powder exactly at the stage of precursor foaming, determining the size of bubbles, wall thickness, the dynamics of reactions, etc.

On the other hand, it is likely that the combustion temperature only correlates with the particle size of the SHS powder and does not completely determine it. The oxidizer–fuel ratio influences, among other things, the viscosity of the precursor melt and its surface-active properties, which can ultimately change the characteristics of the foam more significantly than other factors and can determine the degree of agglomeration of the product particles. For example, in the glycine–nitrate SHS of 50% vol. Y_2_O_3_–50% vol. MgO powders, although there is an extreme dependence of the specific surface area on the precursor composition, it is shifted in the direction of a large fuel excess [22].

As-prepared LiF-doped yttria powders have a minimum specific surface area when obtained from precursor *φ* = 0.375; it is almost three times greater in the studied edge compositions *φ* = 0.3 and *φ* = 0.5. It follows from XRD and SEM data that the sintering activity of such powders is high. Therefore, the SHS reaction temperature has a decisive influence on their size, as opposed to the case of the powders without sintering additive. The specific surface area of powders containing LiF is less by 10–30 m^2^/g. However, the high rate of SHS prevented the processes of crystallization and equilibrium morphology formation from completing normally. Air calcination at 800 °C for 5 h decreased the specific surface area and increased the particle size despite being at a significantly lower temperature than that achieved during SHS. The extreme dependence of the particle size on the precursor composition was retained after calcination, and the equivalent particle size is about four times smaller than that of powders containing no LiF. 

The calcination temperature of the powders was lower than the melting point of LiF (848 °C); nevertheless, the effect of the sintering additive on crystallization is significant. This confirms the XRD data (Figure 2) and the dilatometric results given in [19] that the mechanism of LiF action is not associated with the formation of the liquid phase, providing increased mass transfer as, for example, in LiF-doped MgAl_2_O_4_ powders [38]. Due to the interaction of LiF with Y_2_O_3_, yttrium fluorides and yttrium oxofluorides are formed, which (partially) dissolve in the yttria, disordering its lattice and significantly accelerating the diffusion processes.

### 3.3. Properties of Y_2_O_3_ Ceramics Produced by Hot Pressing of SHS Powders

The chosen conditions for hot pressing (HP) result in a series of visually transparent yttria ceramics for all powders with different precursor compositions, *φ*, from 0.3 to 0.5. However, ceramics made of undoped powders have significantly worse quality. The main reason for this is the interaction of powders with residual carbon-containing gases in the HP chamber. Nonuniform change in the density of the compact during HP provided a different rate of diffusion of residual gases, which was “visualized” in the ceramics as a network of pores penetrating the volume. The distribution of these defects is random, and it is not possible to compare the quality of these samples.

The fluorine-containing additive minimizes the interaction of residual gases with yttria and decreases the amount of associated defects (pores and carbon inclusions). Only on the periphery of the ceramics, where the powder directly contacts the graphite walls of the mold, is the formation of a black rim observed (Figure 5), which has increased porosity. Otherwise, the samples have a homogeneous structure with almost no defects visible to the naked eye. They were used to further study the relationship between SHS precursor composition and ceramic quality.

Figure 6 shows SEM micrographs of the fractured ceramic surfaces. The microstructure of the yttria ceramics is dense; no secondary phase inclusions are observed. Predominantly intragranular fractures in all samples are not typical for ceramics consolidated with fluoride sintering additive. As a rule, there is segregation of residual fluorine-containing impurities at the grain boundaries, which decreases bond strength compared to volume and provides predominantly intergranular ceramic fracture [37,38,39]. This also indirectly confirms that LiF in Y_2_O_3_ does not form a separate phase, but interacts and dissolves in the lattice, at least in the amounts used in the present work. The SEM micrographs show only few submicron-sized pores. This is due not only to the low content of pores, but probably to a greater extent to the fact that the destruction of ceramics occurs through pores and they become difficult to distinguish on fractograms.

In our case, optical microscopy turned out to be more informative in terms of identifying defects that limit the transparency of samples. Individual pores of about 0.5–2 µm in size are detected in the volume of all ceramics. Their number does not differ significantly from sample to sample. There are more pores in the central region of the ceramics, and a single amount is closer to the periphery. Their number is more determined by the HP mode than by the characteristics of the powders used. Such pores usually heal effectively during hot isostatic pressing (HIP). The most significant difference between the samples is the number of pore clusters, an example of which is shown in Figure 7a. They consist of rounded pores located both on the grain boundaries and in the volume, as well as extended pores up to 2 µm in diameter and up to 20–30 µm in length distributed along the grain junctions. Table 1 shows the number of such pore clusters in the central part of the samples on an area with a diameter of 5 mm. It can be seen that their number differs by several times and is the smallest in the ceramics obtained from precursors *φ* = 0.45 and *φ* = 0.5. These pore clusters appear due to the presence of hard agglomerates in the powders. The mode of HP and subsequent HIP can change the size and morphology of such clusters, but it is not possible to completely remove them (see Figure 7b). It is the number of pore clusters that limits the quality of the ceramics, and within the framework of the present work is the criterion for selecting the precursor composition.

The transmission spectra of the ceramics in the visible and infrared regions are shown in Figure 8a,b, respectively. The IR spectra of Y_2_O_3_ ceramics are almost identical; transmittance at a wavelength of 5 µm is 84.5 ± 1.0%, which corresponds to those for the yttria single-crystal in this range. The short-wave absorption edge for all samples is quite sharp, corresponding to a wavelength of 223 nm. At a wavelength of ~300 nm, an absorption band is observed, which is also often present in ceramics of other compositions obtained by HP in graphite molds, for example, in zinc-aluminum spinel [39]. It is most likely caused by penetration of carbon into the samples. This band is present in all ceramics, regardless of the precursor composition; thus, graphite tooling is a more significant source of carbon contamination than SHS with fuel excess. This band disappears after HIP processing (see Figure 8a).

At a wavelength of 1 μm, the samples obtained from precursors close to stoichiometric *φ* = 0.375 and 0.4 have the lowest transmittance of 74.5 % and 75.2 %, respectively, while the samples with *φ* = 0.45 and 0.5 have the best transmittance of 80.1 % and 79.3 %, respectively. Despite the fact that samples *φ* = 0.45 and 0.5 contain fewer pore clusters; their higher transmittance in the visible range is not related to this. A relatively small number of quite large pores (0.5–2 μm) do not contribute to significant losses in transmission measurements in thin (1 mm) samples on spectrometers.

Birefringence is the main source of scattering in HPed ceramics. Inside the, samples individual crystallites are visible (Figure 7a), and transmission-polarizing microscope images clearly show distortions whose scale corresponds to the grain size (Figure 9a). This is most likely due to dissolved fluorides or/and oxofluorides that did not fully evaporate during HP. Their different concentrations or compositions change the refractive indices in adjacent crystallites. Previously, a two-stage HIP was proposed for processing HPed ceramics [19]. The first long stage at moderate temperatures allows the fluorides to evaporate, and the second, high-temperature stage, leads to the healing of pores. If the first step is skipped, the pores cannot heal due to the back-pressure of fluoride in them at high temperature. Figure 9b shows a marked reduction in distortion after the first stage of the HIP at 1200 °C for 20 h, and Figure 9c shows the same after HIP at 1600 °C for 2 h in the ceramics obtained from precursor *φ* = 0.45. Although there are still imperfections, the ceramics can be classified as laser quality.

As a rule, powder milling is an indispensable step in any laser-ceramic technology. It allows the powders to be deagglomerated to a large extent. Less commonly, additional classification of milled powders is also performed. These stages are quite time-consuming, have many nuances, and even with proven technology have a significant risk of powder contamination by impurities and large particles, such as spalls of the balls and grinding jar. In the present work, as-prepared SHS Y_2_O_3_ was used for HP without milling or classification. This made it possible to establish subtle differences in powders with similar characteristics and morphology and to choose the most preferable precursor composition *φ* = 0.45 for SHS. Obviously, such powders cannot be used directly to make large-sized ceramics; they also require milling and classification. Another situation may be with producing thin elements for lasers with a short resonator, such as for microchip lasers In this case, the described technology can be competitive and cost effective; such lasers can show decent efficiency and find practical applications.

## 4. Conclusions

Thermodynamic and experimental studies of self-propagating high-temperature synthesis (SHS) of Y_2_O_3_ nanopowders from precursors of composition *φ*Y(NO_3_)_3_–(1-*φ*)NH_2_CH_2_COOH were performed. The region of stable combustion of precursors is at values of the mole fraction of oxidizer 0.3 ≤ *φ* ≤ 0.5. The addition of lithium fluoride to the SHS precursor increases the crystallinity of Y_2_O_3_ powders and decreases their specific surface area. Powders air annealed for 5 h at 800 °C have an extreme dependence of the particle diameter from *φ*. Maximum d(BET)= 52 and 187 nm for undoped and LiF-doped yttria, respectively, were prepared from a precursor with a stoichiometric oxidizer–fuel ratio *φ* = 0.375. Powders with 1% wt. LiF and precursor composition 0.3 ≤ *φ* ≤ 0.5 were consolidated into optical ceramics by hot pressing (HP) and had high transmittance in the entire transparency range—in the IR range it was almost theoretical and in the visible it was about 90% of theoretical. The main defects limiting transmittance in the visible and near-IR ranges of HPed ceramics are related to birefringence. These defects are practically removed in two-step hot-isostatic pressing, and the quality of the ceramics is mostly determined by the large pores formed from the rigid agglomerates in the powders. The composition of the precursor *φ* = 0.45, which provides the powder with the lowest degree of agglomeration, was determined. Although these powders still require deagglomeration to produce large-sized ceramics, they can be used directly to make high-quality thin discs, such as for microchip lasers.

## Figures and Tables

**Figure 1 materials-16-00260-f001:**
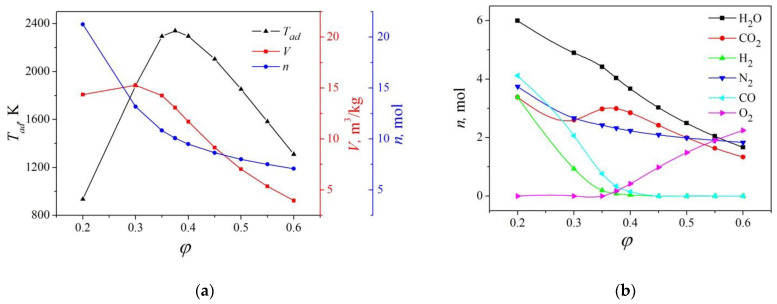
Dependence of (**a**) adiabatic temperature (*T_ad_*), volume (*V*), and number of moles (*n*) of volatile reaction products; (**b**) amount of the main volatile reaction products in the precursor composition *φ*Y(NO_3_)_3_–(1-*φ*)NH_2_CH_2_COOH.

**Figure 2 materials-16-00260-f002:**
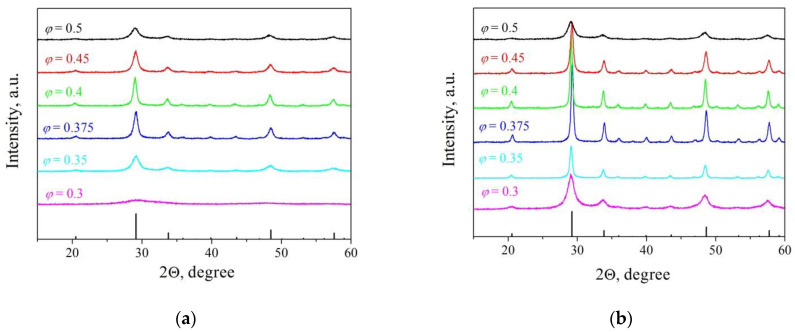
XRD patterns of (**а**) Y_2_O_3_– and (**b**) Y_2_O_3_–LiF nanopowders obtained by self-propagating high-temperature synthesis (SHS).

**Figure 3 materials-16-00260-f003:**
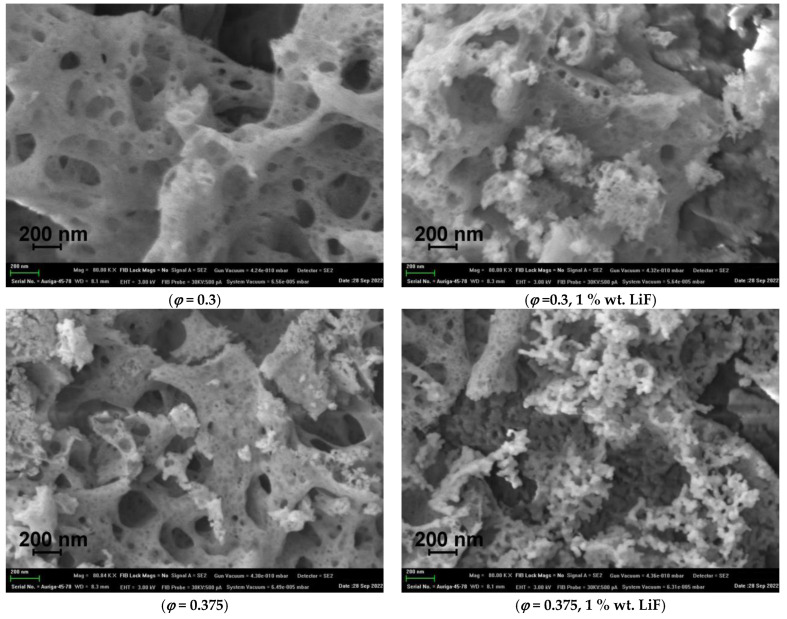
Micrographs of the Y_2_O_3_ powders obtained by the SHS method from precursors with different compositions with and without LiF.

**Figure 4 materials-16-00260-f004:**
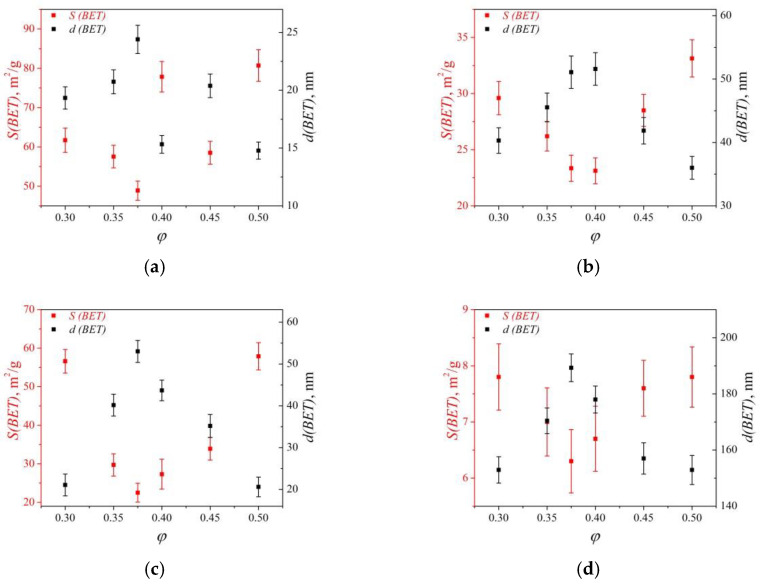
Specific surface area and calculated equivalent diameter of Y_2_O_3_ particles depending on the SHS precursor composition: (**a**,**c**) as-prepared, (**b**,**d**) after air calcination at 800 °C for 5 h, (**a**,**b**) without LiF, and (**c**,**d**) with LiF.

**Figure 5 materials-16-00260-f005:**
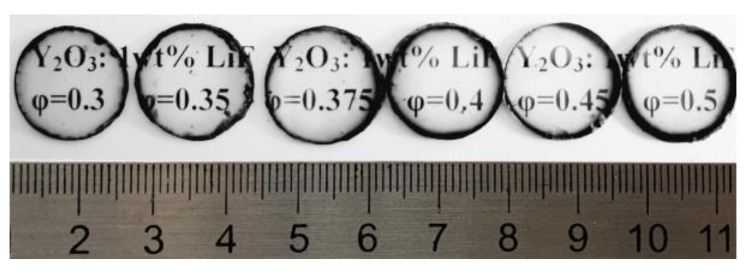
Appearance of the Y_2_O_3_ ceramics with different SHS precursor compositions *φ* and 1wt.% LiF sintering additive.

**Figure 6 materials-16-00260-f006:**
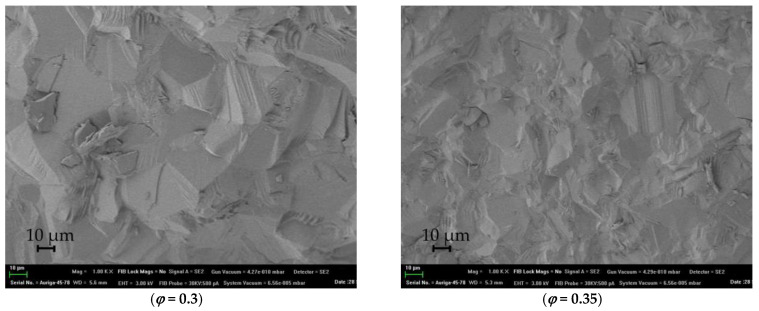
SEM micrographs of Y_2_O_3_ ceramic fractures with different SHS precursor compositions *φ* and 1 wt. % LiF sintering additive.

**Figure 7 materials-16-00260-f007:**
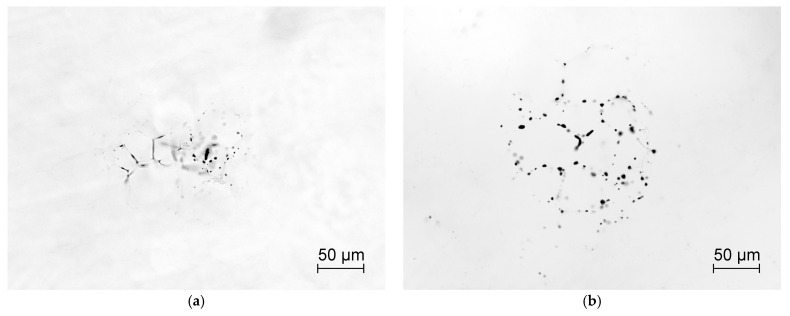
Examples of pore clusters in ceramics (**a**) after hot pressing and (**b**) after HIP.

**Figure 8 materials-16-00260-f008:**
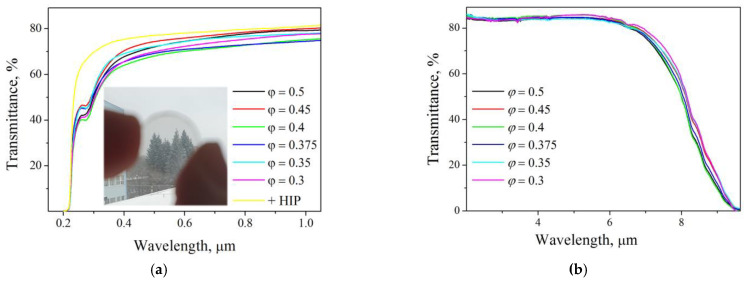
In-line transmission spectra of the Y_2_O_3_ ceramics in the (**a**) visible and (**b**) infrared regions. Outside view seen through the HIPed *φ* = 0.45 ceramics and its transmission spectrum in the visible region are given for comparison.

**Figure 9 materials-16-00260-f009:**
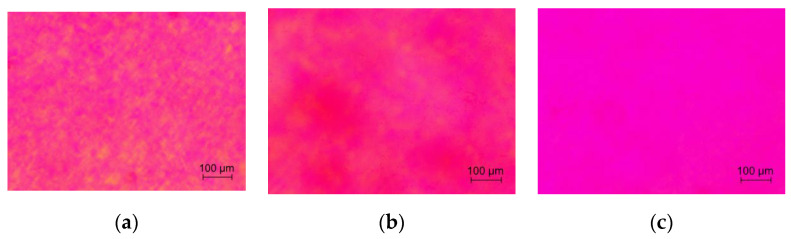
Transmission microscopy photographs with polarizing plates of the Y_2_O_3_ ceramics: (**a**) hot-pressed, (**b**) HIP at 1200 °C for 20 h, and (**c**) HIP at 1600 °C for 2 h.

**Table 1 materials-16-00260-t001:** Number of pore clusters in the area Ø5 mm in yttria ceramics obtained from precursors of different compositions.

Precursor Composition, *φ*.	Number of Pore Clusters
0.3	80
0.35	120
0.375	65
0.4	60
0.45	16
0.5	24

## Data Availability

Not applicable.

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
