# Peer review of "Influence of SHS Precursor Composition on the Properties of Yttria Powders and Optical Ceramics"

_materials, 2022, doi:10.3390/ma16010260_

Round 1
Reviewer 1 Report
This is a good quality work which is recommended to publish almost (apart from some editorial changes) unaltered. E.g. the amendment obviously needed is for the abstract where the statement “The HPed ceramics had transmittance in the range 74.5% to 80.1% @ 1μm at a at a thickness of 1 mm, which was limited by optical inhomogeneity due to incomplete evaporation of the sintering additive…” (line 20-22) is confusing for the reader. Please clearly state what parameters (transmittance or thickness) is limited.
Another question to authors is to explicit reveal (Chapter 3.1) the computer code used for thermodynamic modelling, and whether it has or not the possibility to account the presence of low content (1 wt.%) of LiF. The referring to paper [19] (lines 138-140) does not provide any proof and the word “obviously” cannot serve as proof. As example the computer code ASTRA with such possibility shows that small additives can have important effects – see e.g. DOI: https://doi.org/10.1557/PROC-757-II3.13.
Author Response
Point 1: The amendment obviously needed is for the abstract where the statement “The HPed ceramics had transmittance in the range 74.5% to 80.1% @ 1μm at a at a thickness of 1 mm, which was limited by optical inhomogeneity due to incomplete evaporation of the sintering additive…” (line 20-22) is confusing for the reader. Please clearly state what parameters (transmittance or thickness) is limited.
Response 1: Thank you for the suggestion. We have made changes so as not to confuse the reader.
Changes in the text: The 1 mm thick HPed ceramics had transmittance in the range 74.5% to 80.1% @ 1μm, which was limited by optical inhomogeneity due to incomplete evaporation of the sintering additive.
Point 2: Another question to authors is to explicit reveal (Chapter 3.1) the computer code used for thermodynamic modelling, and whether it has or not the possibility to account the presence of low content (1 wt.%) of LiF. The referring to paper [19] (lines 138-140) does not provide any proof and the word “obviously” cannot serve as proof. As example the computer code ASTRA with such possibility shows that small additives can have important effects – see e.g. DOI: https://doi.org/10.1557/PROC-757-II3.13.
Response 2: Yes, the computer code used allows to take into account the presence of LiF. As one of the most thermodynamically stable compound in the considered system, LiF practically does not participate in reactions. Its influence is limited to the decrease of the adiabatic temperature due to its heating and evaporation (approximately by 50-100 K depending on the precursor composition), keeping the appearance of the curves unchanged. We optimized the amount of LiF introduced for the specific vacuum level, hot pressing mode, powder dispersion, mold design, etc. We believe that other researchers will have to perform their own concentration optimization to reproduce the results. Therefore, it seems correct to give calculation curves without LiF. We have made some changes in the text to make our viewpoint clear.
Changes in the text:
According to the calculations LiF is thermodynamically stable compound in the considered system and practically does not participate in reactions (partly due to the lack of reliable data on possible products of the chemical interaction of Y2O3 and LiF). Its influence is limited by the reduction of the adiabatic temperature due to its heating and evaporation (approximately by 50-100 K depending on the precursor composition). Therefore it cannot alter the general results of thermodynamic calculations; nevertheless, its effect on the morphology of powders is significant due to changes in the diffusion activity of the components [19].
Thank you for comments and appreciation of the work.
Reviewer 2 Report
The manuscript elaborated SHSed method for the preparation of Y2O3 nanopowders via thermodynamic and experimental studies. The result of this work is good and contains considerable interests to readers. I fully recommend its publication after the authors address the following issues.
(1) In “Materials and Methods” section, the authors forgot to describe the subsequent HIP process after HP.
(2) The black rims on the sintered ceramic samples could be observed. Why not remove them by annealing in oxygen or air.
(3) P. 8: The authors claim, “The fluorine-containing additive minimizes the interaction of residual gases with yttria, and decreases the amount of associated defects”. What kinds of defects do you mean here? I think microdefects are correct, but point defects are improper because the relatively small ionic radius for Li+ may dissolve into the Y2O3 lattice to create point defects via the formation of interstitial solid solution like Si4+. Two references (Doi: 10.1016/j.jeurceramsoc.2020.07.029; Doi: 10.1016/j.ceramint.2019.04.203) may be helpful for you to reveal the mechanism of LiF additive.
(4) P. 10: The authors claim, “This band 355 disappears after HIP processing”. Why not provide some evidences? For example, show the transmittance curves of HIPed Y2O3 ceramics.
(5) Minor grammar mistakes should be avoided.
In conclusion, I recommend its publication after minor revision.
Author Response
Point 1: In “Materials and Methods” section, the authors forgot to describe the subsequent HIP process after HP.
Response 1: Thank you for the observation. A description of the HIP process has been added.
Changes in the text: Hot isostatic pressing was performed on the UGL-2000 plant (Russia) in a graphite crucible at 1200 °C, 100 MPa for 20 h, then on the AIP8-30H/WF plant (USA) in an yttria crucible at 1600 °C, 200 MPa for 2 h. The ceramics were then polished again.
Point 2: The black rims on the sintered ceramic samples could be observed. Why not remove them by annealing in oxygen or air.
Response 2: To remove black rims, an annealing at a relatively high temperature is required. Since hot pressing is performed at a low vacuum, some residual gases are trapped in the pores. Annealing leads to pore expansion and disturbance of the ceramic structure. Therefore, we avoid it. During the HIPing process, the black rims disappear. We added a photo of the HIPed sample (it was also not annealed in air) to the transmission spectra.
Point 3: P. 8: The authors claim, “The fluorine-containing additive minimizes the interaction of residual gases with yttria, and decreases the amount of associated defects”. What kinds of defects do you mean here? I think microdefects are correct, but point defects are improper because the relatively small ionic radius for Li+ may dissolve into the Y2O3 lattice to create point defects via the formation of interstitial solid solution like Si4+. Two references (Doi: 10.1016/j.jeurceramsoc.2020.07.029; Doi: 10.1016/j.ceramint.2019.04.203) may be helpful for you to reveal the mechanism of LiF additive.
Response 3: Yes, one of the main reasons for adding LiF is to create point defects. We have corrected the description so as not to confuse the reader.
Changes in the text: The fluorine-containing additive minimizes the interaction of residual gases with yttria, and decreases the amount of associated defects – pores and carbon inclusions.
Point 4: P. 10: The authors claim, “This band 355 disappears after HIP processing”. Why not provide some evidences? For example, show the transmittance curves of HIPed Y2O3 ceramics.
Response 4: Thank you for the suggestion. We have added transmittance curve of HIPed Y2O3 ceramics.
Point 5: Minor grammar mistakes should be avoided.
Response 5: Grammatical mistakes have been corrected.
Thank you for comments and appreciation of the work.
Reviewer 3 Report
The present manuscript deals with a new procedure SHS relevant for the sintering process. The fabrication of optical ceramics is a major issue at this moment and the process for yttria powders is lear explained and convincing, so other labs interested can repeat the sintering and check the experimental results.
The microanalysis of the resulting ceramics is well explained, as well as the optical properties, particularly their transmittance and birefringency. The transmittance is high but an even higher value would have been desiderable, this is the reason why I rated the scientific soundness as average.
Optionally, but I miss a comment or measure related to the damage thresholds of the different samples. To be convinced of the applicability of those optical ceramics for laser microchips, as authors indicate in the conclusions, an evaluation of the damage thresholds would have been expected.
Author Response
Point 1: Optionally, but I miss a comment or measure related to the damage thresholds of the different samples. To be convinced of the applicability of those optical ceramics for laser microchips, as authors indicate in the conclusions, an evaluation of the damage thresholds would have been expected.
Response 1: Previously, we obtained laser generation on ceramics made from SHS powders. In particular, a 64.5% slope efficiency was achieved on Yb:Y2O3 continuous-wave micro-laser on a ceramics fabricated by a method described in this work [doi:10.1016/j.optmat.2021.111349].
The main goal of this work was to optimize the precursor composition for SHS yttria. We agree with the utility of the damage threshold measurements, but we believe that they should be the subject of a separate work. The damage thresholds are highly dependent on other technological parameters (purity of reagents, heterogeneous inclusions, HP and HIP modes, polishing quality, etc.), it is unlikely that we can isolate the effect of precursor composition on their background.
Thank you for comments and appreciation of the work.